# Effect of high glucose on the gene expression profiling in cardiac fibroblasts from rats at different ages

Quqian Mo[1,*], Angyu Zhan[1,*], Ruining Bai[1], Shaoling Lin[1], Jiaojiao Feng[1], Tongjun Li[1], Zijian Lao[2,3], Xiao Yang[4], Keke Wang[2,3], Xianglu Rong[1] and Lexun Wang[1]

[1] Guangdong Metabolic Diseases Research Center of Integrated Chinese and Western Medicine, Guangdong Pharmaceutical University, Guangzhou, Guangdong Province, China
[2] Department of Emergency, The First Affiliated Hospital of Sun Yat-sen University, Guangzhou, Guangdong Province, China
[3] NHC Key Laboratory of Assisted Circulation and Vascular Diseases, Sun Yat-sen University, Guangzhou, Guangdong Province, China
[4] Department of Clinical Laboratory, Guangzhou First People's Hospital, School of Medicine, South China University of Technology, Guangzhou, Guangdong Province, China
[*] These authors contributed equally to this work.

## ABSTRACT

**Background**. Cardiac fibroblasts (CFs) play a vital role in the physiological and pathological processes of the heart. Previous studies have demonstrated that high glucose stimulation induces the transformation of CFs into myofibroblasts, contributing to cardiac fibrogenesis. However, *in vivo* experiments have predominantly utilized adult animals, whereas most *in vitro* studies have focused on CFs derived from neonatal animals. The responses of CFs from different age groups to high glucose levels remain unclear. This study aimed to investigate transcriptional alterations in CFs at distinct developmental stages in response to high glucose exposure.

**Methods**. CFs were isolated from neonatal (S1, 0–3 days), juvenile (S2, 3–4 weeks), adult (S3, 10–13 weeks), and aged (S4, 20 months) rats. CFs were exposed to normal (5.5 mM, NG) or high glucose (33 mM, HG). The cellular RNA was extracted for sequencing and analysis. Differentially expressed genes (DEGs) were validated by quantitative real-time PCR.

**Results**. After NG treatment, fibrosis and inflammation-related gene expression in CFs (*e.g.*, *Col8a1*, *Col8a2*, *IL-6*, *Ccl2*, *Ccl20*, *Mmp2* and *Mmp9*) increased with age, while proliferation-related genes (MCM family, *Sox10*, *Sox11*) decreased. HG treatment most affected S3-CFs, showing 228 DEGs; it suppressed growth-related genes (*Adra1d*, *Htr2b*) and enhanced inflammatory genes (*IL-6*, *Olr1*). In S1-CFs, 197 inflammation-rich genes were upregulated under HG. S4-CFs displayed 166 DEGs, mostly metabolic downregulation (*G6pc*). S2-CFs had the fewest DEGs (112), focusing on cell metabolism.

**Conclusions**. Fibrosis- and inflammation-associated gene expression in CFs showed an age-dependent stepwise elevation. CFs from distinct developmental stages responded differently to HG stimulation, with S3-CFs exhibiting the most pronounced response. These findings highlight the developmental characteristics of CFs and provide implications for the selection of appropriate CFs to investigate diabetes-associated cardiac fibrosis.

Corresponding authors
Xianglu Rong, xlrong@gdpu.edu.cn
Lexun Wang,
wanglexun123456@163.com

## INTRODUCTION

Diabetes mellitus is an independent risk factor for cardiovascular disorders. Compared with non-diabetic individuals, diabetic patients face a 2–4-fold increased risk for coronary artery disease and a 4–6-fold increased risk of myocardial infarction and congestive heart failure, respectively (*Prandi et al., 2023*). Diabetes causes abnormalities in cardiac metabolism, structure, and function. Hyperglycemia, insulin resistance, and hyperlipidemia in the diabetic milieu contribute to advanced glycation end-product formation, oxidative stress, inflammation, and abnormal cytokine profiles, collectively resulting in cardiac remodeling and contractile dysfunction (*Prandi et al., 2023*; *Tan et al., 2020*). Diabetic heart disease encompasses coronary artery disease, cardiac autonomic neuropathy, and diabetic cardiomyopathy (DCM) (*Prandi et al., 2023*). Interstitial and perivascular fibrosis are prominent features of all forms of diabetic heart disease and can lead to heart failure (*Tuleta & Frangogiannis, 2021a*; *Tuleta & Frangogiannis, 2021b*). Current therapies using hypoglycemic and hypolipidemic agents may partially reduce myocardial collagen deposition; however, effective measures to prevent DCM are lacking due to the unclear mechanisms underlying diabetes-induced myocardial fibrosis.

Cardiac fibroblasts (CFs) play a crucial role in the pathogenesis of myocardial fibrosis. Following cardiac injury, CFs become activated into myofibroblasts, releasing pro-inflammatory cytokines and fibrogenic growth factors that promote cardiac fibrotic remodeling (*Travers et al., 2016*). Moreover, the extant literature has shown that high glucose (HG) levels can stimulate CFs to overproduce extracellular matrix constituents, exacerbating myocardial fibrosis (*Liu, López De Juan Abad & Cheng, 2021*). However, most *in vivo* models of diabetic heart disease have utilized adult animals, whereas *in vitro* experiments have primarily employed CFs derived from neonatal rats (*Ding et al., 2021*; *Lin et al., 2021*; *Liu et al., 2020*; *Zong et al., 2020*). Given that neonatal rat hearts can fully regenerate after injury, unlike the reparative scarring observed in adults (*Bassat et al., 2017*; *Nishiyama et al., 2022*; *Porrello et al., 2011*), we hypothesized that CFs from different age groups would demonstrate distinct gene expression patterns and responses to stimuli. However, gene expression profiles and HG responses of primary CFs across various age groups remain unclear.

In this study, we isolated primary CFs from juvenile and adult rats, treated them with normal glucose (NG, 5.5 mM) or HG (33 mM) for 24 h, and analyzed differentially expressed genes (DEGs) using RNA sequencing and quantitative real-time PCR (Fig. 1), aiming to characterize the age-dependent genomic and functional traits of CFs.

## MATERIALS & METHODS

### Animals

Male Sprague-Dawley (SD) rats at 0–3 days (Stage 1, S1), 3–4 weeks (Stage 2, S2), 10–13 weeks (Stage 3, S3) and 20 months (Stage 4, S4) were purchased from Guangdong Medical

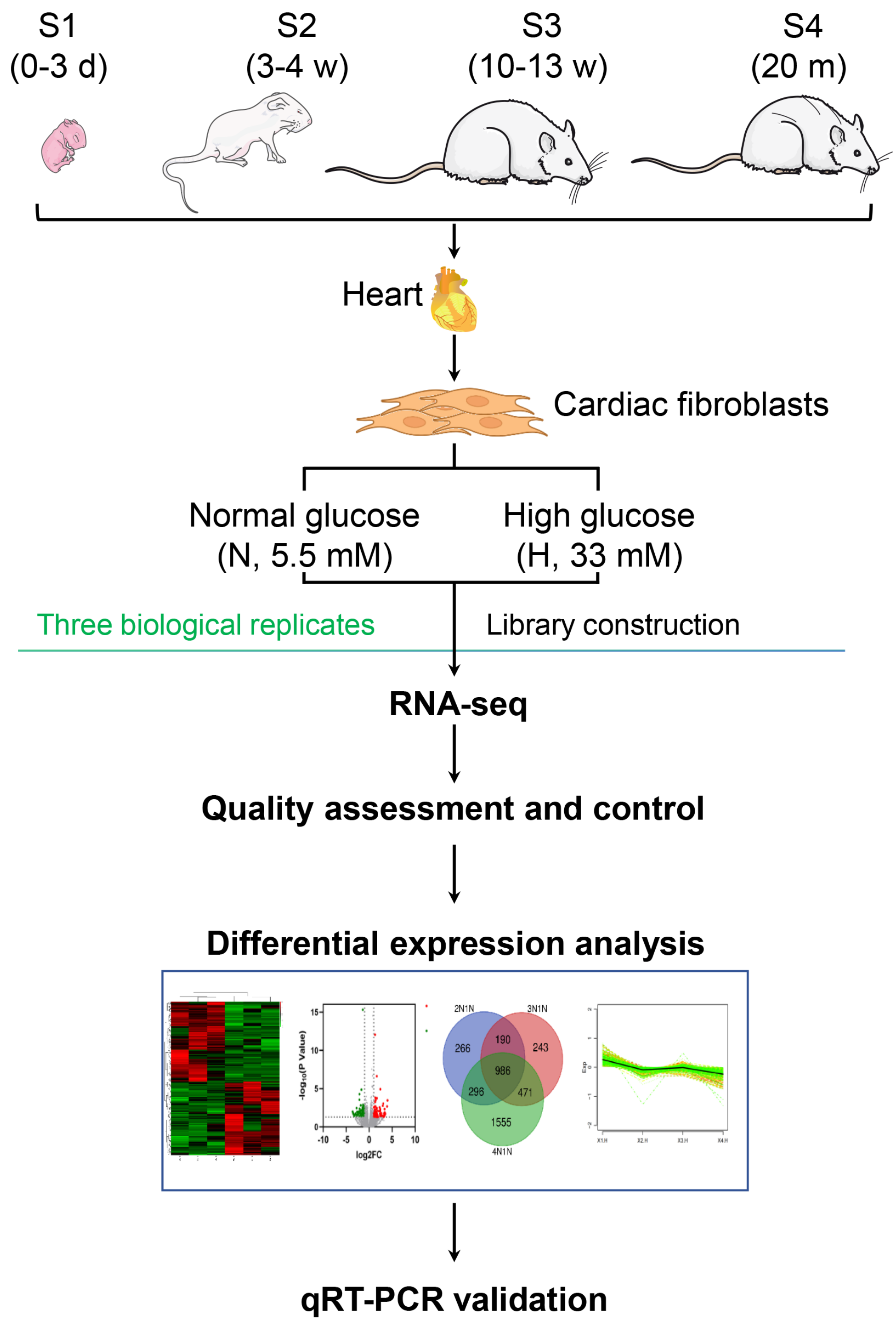

**Figure 1** Flow chart of comparative transcriptome analysis.

Laboratory Animal Center. All rats were housed in individual cages in SPF environments with free access to water and food, provided with a 12/12 h light/dark cycle, maintained at a temperature of 25 °C and a relative humidity of 50%. The experiment began after one week of acclimatization feeding. Rats were anesthetized with tribromoethanol for euthanasia, inducing unconsciousness in the experimental animals in the shortest possible time to minimize pain and fear. The chest was opened to remove the heart, and CFs were isolated for culture. In this study, no rats were euthanized before the end of the experimental plan. This study was approved by the Animal Care and Ethics Committee of Guangdong Pharmaceutical University (approval number: gpdulac17), and all procedures were performed in accordance with the Guide for the Care and Use of Laboratory Animals (2011) (*National Research Council Committee for the Update of the Guide for the & Use of Laboratory, 2011*).

## Isolation of primary cardiac fibroblasts

CFs were isolated from hearts of S1, S2, S3 and S4 rats separately.

For S1 rats, the ventricles were minced and first digested with 0.125% trypsin for 10 min followed by collagenase II with repeated digestions every 10 min until completely digested. The cell suspension was collected and resuspended in DMEM containing 10% fetal bovine serum (FBS). After centrifugation, cells were plated in DMEM with 5.5 mM glucose, 10% FBS (10099141; Gibco, Waltham, MA, USA), and 1% penicillin/streptomycin (15140122; Gibco, Waltham, MA, USA) and incubated at 37 °C in 5% $CO_2$ for 1.5 h. Non-adherent cells were then washed off and the culture medium was replaced (*Wang et al., 2017a*). The S1 sequencing samples include three biological replicates, each of which consists of CFs extracted from the hearts of different rats.

For S2, S3 and S4 rats, the chest cavity was opened under tribromoethanol (LOT # MKCM8927; Sigma, St. Louis, MO, USA) anesthesia to expose the heart. The heart was immediately flushed with 10 mL EDTA buffer injected into the right ventricle. The ascending aorta was clamped with a hemostat under sterile conditions and the heart quickly transferred to a 60 mm culture dish containing fresh and cold EDTA buffer. 10 mL EDTA buffer and 50 mL collagenase buffer (0.08% collagenase II and 0.25% trypsin, Worthington, Cat #: LS004176; 25200056; Gibco, Waltham, MA, USA) were sequentially injected into the left ventricle for digestion. The atria and ventricles were then separated and cut into small pieces (one $mm^3$) using scissors. The minced tissue was placed in 50 mL sterile tubes containing collagenase buffer and shaken at 37 °C for 15 min. The supernatant was collected into 50 mL tubes, and digestion was terminated by adding five mL FBS. After centrifugation, cells were resuspended in DMEM containing 5.5 mM glucose, 10% FBS, and 1% penicillin/streptomycin and incubated at 37 °C in 5% $CO_2$ for 1.5 h. Non-adherent cells were then washed off and the media was changed. All studies were performed with CFs at passage 3 (*Ackers-Johnson et al., 2016*; *Wang et al., 2017b*). Similarly, the S2, S3, and S4 sequencing samples each contain three biological replicates, with each biological replicate being CFs extracted from the hearts of different rats.

### Treatment

The four groups of CFs were seeded at a density of $3 \times 10^5$ cells/well in six-well plates and cultured in DMEM containing NG (5.5 mM) or HG (33 mM) for 24 h (*Guo et al., 2018*). Cellular RNA was then extracted (three biological replicates for NG and HG among S1, S2, S3 and S4, 24 samples in total).

### Library construction and RNA sequencing

Total RNA was extracted using the RNeasy Mini Kit (Qiagen, Hilden, Germany). Following the manufacturer's protocol, paired-end libraries were synthesized using the TruSeq RNA Sample Preparation Kit (Illumina, San Diego, CA, USA) (*Carmona-Mora et al., 2023*). In brief, poly-A mRNA molecules were purified using poly-T oligo-attached magnetic beads. Purified libraries were quantified by Qubit 2.0 Fluorometer (Life Technologies, Gaithersburg, MD, USA) and validated on an Agilent 2100 Bioanalyzer (Agilent Technologies, Santa Clara, CA, USA) to confirm insert size and calculate molar concentration. Gene expression profiles were generated on cBot, with libraries diluted to 10 pM and sequenced on Illumina HiSeq X-ten (Illumina, San Diego, CA, USA). Library construction and sequencing were performed at Shanghai Biotechnology Corporation. All raw data were submitted to the NCBI Gene Expression Omnibus (GEO) database (GEO accession number: GSE269896).

### Gene expression data analysis

For data analysis, raw reads were filtered by Seqtk before mapping to the genome using Tophat (version: 2.0.9). Gene fragments were counted by HTSeq. Hisat2 (version: 2.0.4) was used to map filtered reads to the *Rattus norvegicus* reference genome. After genomic localization, Stringtie (version: 1.3.0) was run with reference annotations to generate fragment counts per million map fragments (FPKM) values for known gene models. DEGs were identified using edgeR. Significance thresholds for $P$-values were set by false discovery rate (FDR) across multiple tests. Fold changes (FC) were also estimated from FPKM per kilobase of exon model according to each sample. DEGs were selected using the following filtering criteria: $P$-value < 0.05 and $\log_2$FC > 2. Analyses were performed using GraphPad Prism and the "Weishengxin" platform (http://www.bioinformatics.com.cn).

### Principal component analysis and power analysis calculation

Principal component analysis (PCA) was conducted by R (version 2.15.3) using the prcomp (data, center = T, scale. = T) and visualized by ggplot2 package.

The statistical power of this experimental design, calculated in RNASeqPower was 0.92. Three biological replicates were used to achieve the claimed statistical power. No technical repetition was used in the RNA-seq experiment.

### Venn diagram

Shared genes across the three data sets were identified through formulating a Venn diagram. Using the Venn diagram function on the Weishengxin platform, the gene sets for comparison were intersected to form a Venn diagram with the numbers representing quantities of genes.

### Functional enrichment analysis

Gene Ontology (GO) annotation and Kyoto Encyclopedia of Genes and Genomes (KEGG) pathway enrichment analysis were conducted for the DEGs. GO functional annotation and KEGG pathway enrichment analytics were executed utilizing the "Weishengxin" platform. The gene designations and $\log_2$FC values were uploaded into the website to enable enrichment quantification. The threshold denoting significant enrichment was defined as $P$-value < 0.05.

### Cluster analysis and gene annotation

To correct for potential scale effects on gene expression and avoid use of negative expression values, we added 1 to the fragments per kilobase of transcript per million mapped reads (FPKM) values for expressed genes and then $\log_2$-transformed the modified FRKM values. Average expression values across developmental stages were computed for each expressed gene. Data points with undetected expression were replaced with the mean values for pattern comparison. Deviations of each expressed gene from the mean expression level across the four stages CFs under condition of NG treatment were calculated. The temporal expression change patterns for individual genes were clustered using k-means under condition of NG treatment. All statistical analyses were performed using R (version 2.15.3). Clustering of transcript expression profiles based on FPKM levels was performed using the k-means approach and Euclidean similarity metrics (*Guo et al., 2016*).

### Real time-PCR

Total RNA was extracted from cells using Total RNA Extraction Kit (Cat.# LS1040, Promega, Shanghai, China) and cDNA was synthesized using the PrimeScript™ RT reagent Kit (Takara, Tokyo, Japan). PCR primers were designed and synthesized by Sangon Biotech Co., Ltd. (see Table 1). Real-time PCR analysis was performed using TB Green Premix Ex Taq (Takara, Tokyo, Japan) and the LightCycle 480 (Roche, Basel, Switzerland). Gapdh was used as an internal reference standard.

### Statistical analysis

Data are expressed as mean ± standard error of mean (SEM). GraphPad Prism 8 (GraphPad Software, La Jolla, CA, USA) was used for data analysis. $T$-tests and Mann–Whitney test were used for statistical analysis, and $P$-value <0.05 was considered statistically significant.

## RESULTS

### Gene expression differences in primary CFs from different developmental stages

We initially analyzed the gene expression profiles of primary CFs from rats at four developmental stages under NG treatment. As shown in Fig. 2A, PCA showed significant differences among the groups. PC1 and PC2 contributed 53.7% and 18.4% of the variance, respectively. Along the PC2 axis, S1 and S4 showed a positive distribution, while S2 and S3 were negatively distributed (Fig. 2A). S1-CFs differed significantly from the other three groups (Figs. 2A, 2B); therefore, S1-CFs were used as the control group. Compared to S1-CFs, we found 1,736 DEGs in S2-CFs, 1,881 DEGs in S3-CFs, and 3,292 DEGs in S4-CFs.

**Table 1** Primer sequences.

| Primer name | Sequence (5′to 3′) |
|---|---|
| Il6-F | ctt cca gcc agt tgc ctt ctt g |
| Il6-R | tgg tct gtt gtg ggt ggt atc c |
| Aldh3a1-F | ttt gct ggc tgt ctt gtc ctt gag |
| Aldh3a1-R | gtg gaa ttt gga gga gtg agg tga g |
| Ccl2-F | atc acc tgc tac tca ttc act g |
| Ccl2-R | ctt ctt tgg gac acc tgc tgc tg |
| Ccl20-F | gga cac agc cca agg agg aaa tg |
| Ccl20-R | gga caa gac cac tgg gac aca aat c |
| Cxcl1-F | gca gac agt ggc agg gat tca c |
| Cxcl1-R | tga gtg tgg cta tga ctt cgg ttt g |
| Cxcl3-F | tgc ttc tgc tgc ttc tgc tga tg |
| Cxcl3-R | cac cgt caa gct ctg gat gtt ctc |
| Atp1b2-F | agg tgg ttg agg agt gga agg ag |
| Atp1b2-R | acg agg tag aag agg agg atg aag g |
| Lef1-F | cgg caa tcg cag agg ctc ttg |
| Lef1-R | aca ctc gga gac agc agg aag g |
| Dbd2-F | aat gtg agt cgt gct gcc ttc tg |
| Dbd2-R | ttc tgg aga ccc tgc tgt agt gac |
| Erbb3-F | tta gag gag gag gac ggc aat gg |
| Erbb3-R | aca gaa ctg aga cct acc gac gag |
| Prkcq-F | agg acc cac taa ccc gca tcg |
| Prkcq-R | tca cat ccc ttt ccc tcc ctt ctg |
| Gapdh-F | ctc ctc caa ggt cat cca tga caa ct |
| Gapdh-R | aac aaa ggg gcc atc cac agt ctt |

Among the 6,909 DEGs, 1,825 genes were upregulated, and 5,084 were downregulated (Fig. 2C). A Venn diagram of these DEGs showed that 968 genes were altered in all three comparison groups (Fig. 2D). Further analysis of these 968 commonly altered DEGs revealed that 167 were upregulated and 802 were downregulated (Fig. 3A).

### KEGG and GO analysis of upregulated DEGs

We conducted KEGG and GO analyses of the upregulated and downregulated DEGs. KEGG analysis indicated that the upregulated DEGs were primarily involved in longevity-regulating, tyrosine metabolism, and drug metabolism (Fig. 3B), which are primarily related to cellular metabolism. GO enrichment analysis showed that these upregulated DEGs were predominantly involved in biological processes (45.61%), cellular components (28.07%), and molecular functions (26.32%) (Fig. 3C). Specifically, 111 DEGs were associated with cellular process, 111 with single-organism process, and 84 with biological regulation (Fig. 3C). Regarding cellular components, 103 DEGs were related to cell and cell part, and 76 DEGs each were associated with organelle and membrane part (Fig. 3C). In terms of molecular functions, 101 DEGs were involved in binding and 37 in catalytic activity (Fig. 3C).

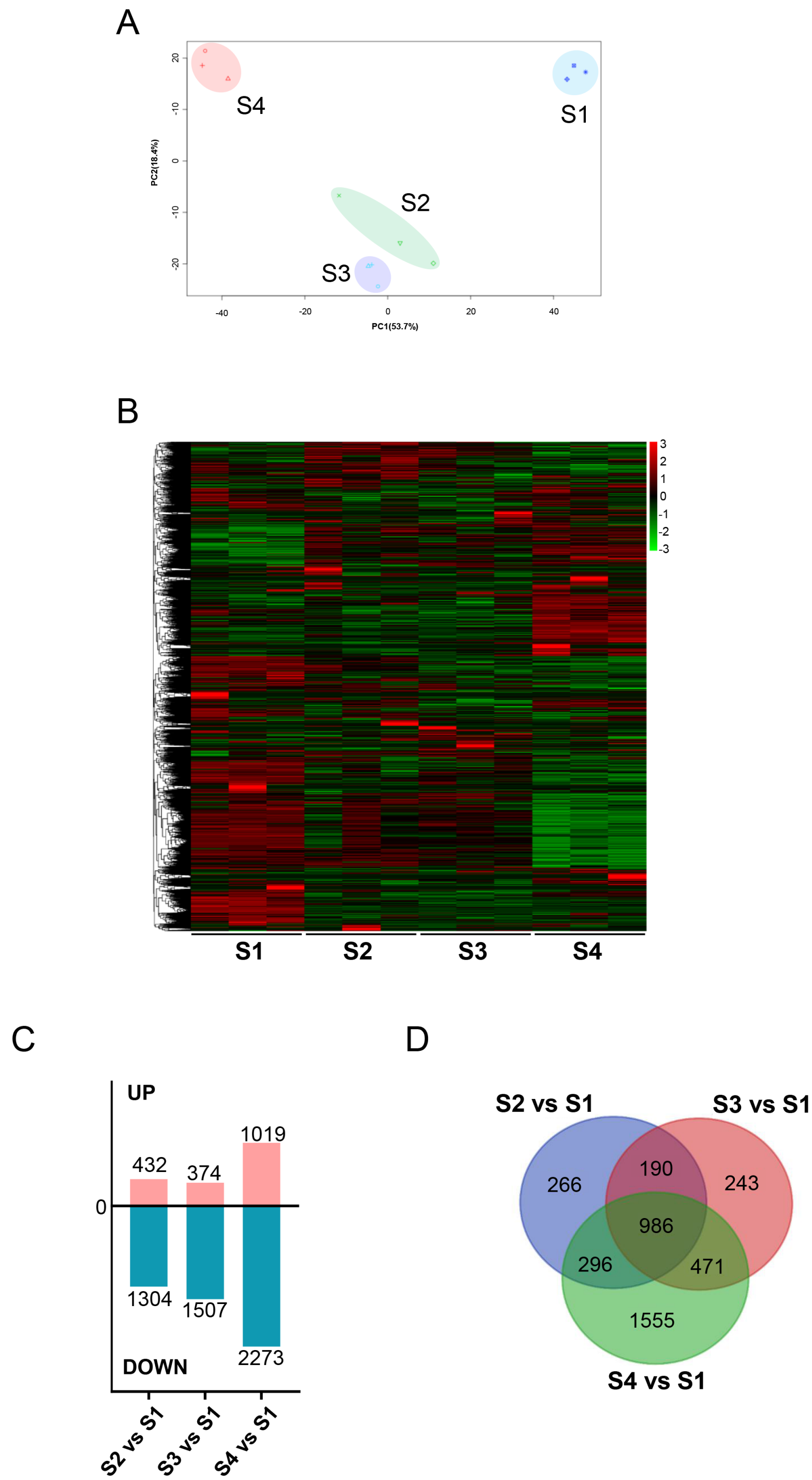

**Figure 2 Comparisons between the four timepoints after N treatment.** (A) PCA plot: displays correlations between samples. (B) Heatmap of DEGs across the four timepoints. (C) Statistics of upregulated and downregulated DEGs in the different comparisons. (D) Venn diagrams: the numbers in each circle indicate DEGs in each timepoint, with overlapping regions denoting commonly expressed genes.

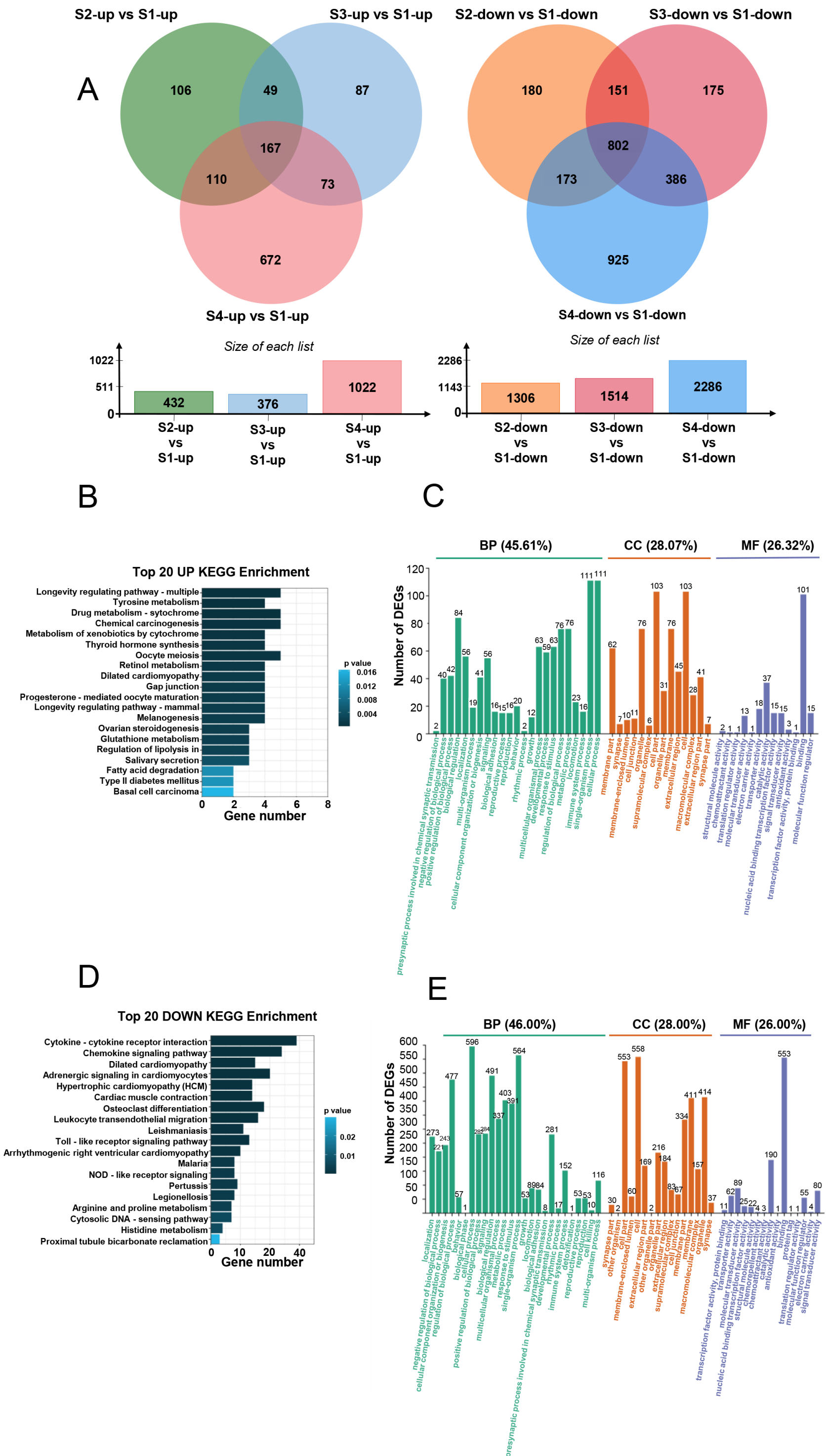

**Figure 3 Analysis of DEGs across the four timepoints after N treatment.** (A) Venn diagrams of upregulated and downregulated DEGs, with overlaps representing commonly expressed genes. (B) Top 20 enriched KEGG pathways for upregulated DEGs. (C) GO enrichment analysis of upregulated DEGs. (D) Top 20 enriched KEGG pathways for downregulated DEGs. (E) GO enrichment analysis of downregulated DEGs.

### KEGG and GO analysis of downregulated DEGs

KEGG analysis of the downregulated DEGs showed their prominent involvement in cytokine-cytokine receptor interaction, chemokine signaling pathway, dilated cardiomyopathy, adrenergic signaling in cardiomyocytes, hypertrophic cardiomyopathy, and cardiac muscle contraction (Fig. 3D). GO enrichment analysis revealed that downregulated DEGs were mainly involved in biological processes (46%) (Fig. 3E), with additional contributions to cellular components (28%) and molecular functions (26%) (Fig. 3E). Specifically, 596 DEGs were involved in cellular process, 564 in single-organism process, and 491 in biological regulation (Fig. 3E). For cellular components, 558 DEGs were associated with cell, 553 with cell part, 414 with organelle, and 411 with membrane (Fig. 3E). Regarding molecular functions, 553 DEGs were involved in binding and 190 in catalytic activity (Fig. 3E).

### Cluster analysis of DEGs

We compared the expression patterns of the DEGs among the four CF stages using k-means clustering, revealing 10 clusters (Fig. 4). In clusters 2 and 10, the DEGs showed a continuous upward trend with increasing age, whereas DEGs in clusters 3, 5, and 7, displayed a continuous downward trend with age (Fig. 4). DEGs in clusters 1 and 6 initially decreased and then increased over time, whereas those in clusters 4, 8, and 9 first increased and then decreased over time (Fig. 4).

Further analysis of DEGs in clusters 2 and 10 revealed 909 upregulated DEGs, including collagen family members *Col8a1*, *Col9a3*, *Col10a1*, *Col11a1*, *Col11a2*, *Col19a1* and *Col20a1*; matrix metalloproteinase gene *Mmp2*; and chemokines *Cxcl13*, *Cxcl14* and *Ccl19* (Fig. 4). These genes are directly or indirectly involved in the fibrotic processes. In clusters 3, 5, and 7, 1,602 DEGs were down-regulated, implicated in wound healing, tissue remodeling (matrix metalloproteinases *Mmp7*, *Mmp8*, and *Mmp12*), mediation of cell proliferation and anti-apoptosis (*IL-7* and *IL-7R*), regulation of cell differentiation, growth and apoptosis (*MapK13* and *Map4K1*) and DNA replication and cell cycle maintenance (*Mcm2*, *Mcm4*, *Mcm5*, *Mcm6*, *Mcm7* and *Mcm8*) (Fig. 4). These findings suggest that with increasing age, the proliferative capacity of CFs decreases, whereas the expression of fibrosis and inflammatory factors increases.

Additionally, the clustering results identified the fibrosis-related gene *Col3a1* in cluster 9.

## Effects of HG on CFs

Subsequently, we stimulated CFs at different stages with HG to characterize their responses. The results revealed that under HG treatment, 197, 112, 228, and 166 DEGs were identified in S1-CFs, S2-CFs, S3-CFs, and S4-CFs, respectively (Fig. 5A). Among the 703 DEGs, 274 were upregulated, and 429 were downregulated (Fig. 5A). Further enrichment analysis of the 274 upregulated DEGs revealed 17 genes commonly present in two or more groups, whereas no commonly upregulated DEGs were identified across the four CFs stages (Fig. 5B, Table 2). Additional examinations revealed that these DEGs were highly pertinent to inflammation (Table 2), indicating that inflammatory genes were initially activated in CFs following the HG challenge.

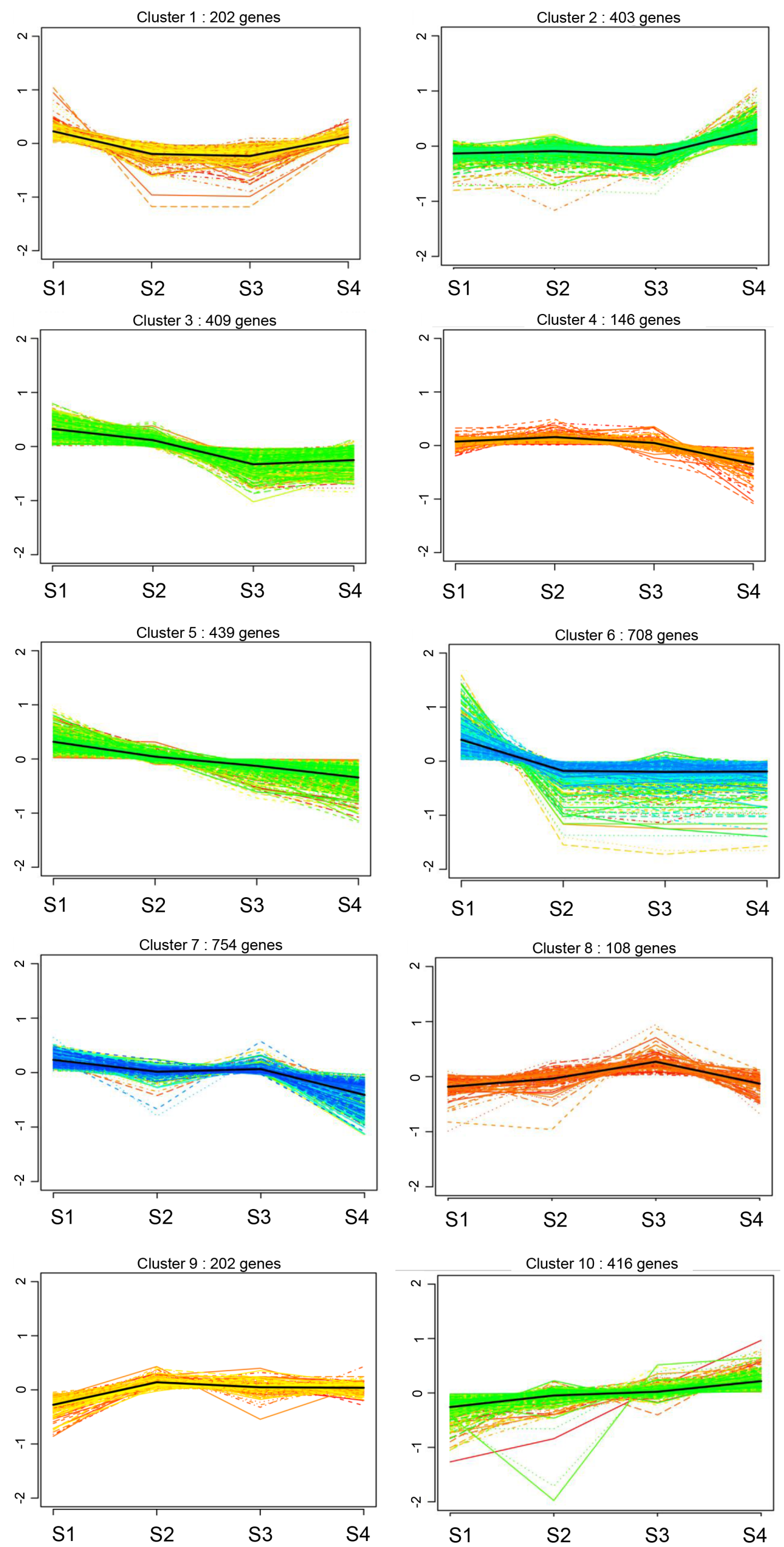

**Figure 4 Cluster analysis of gene expression profiles of rat CFs of different ages under normal glucose conditions.** Clustering was performed by K-means system, with the number of genes in each cluster shown in parentheses. The *y*-axis indicates the deviation of gene expression levels across different ages from the mean expression of genes in the cluster. The *x*-axis denotes the four timepoints.

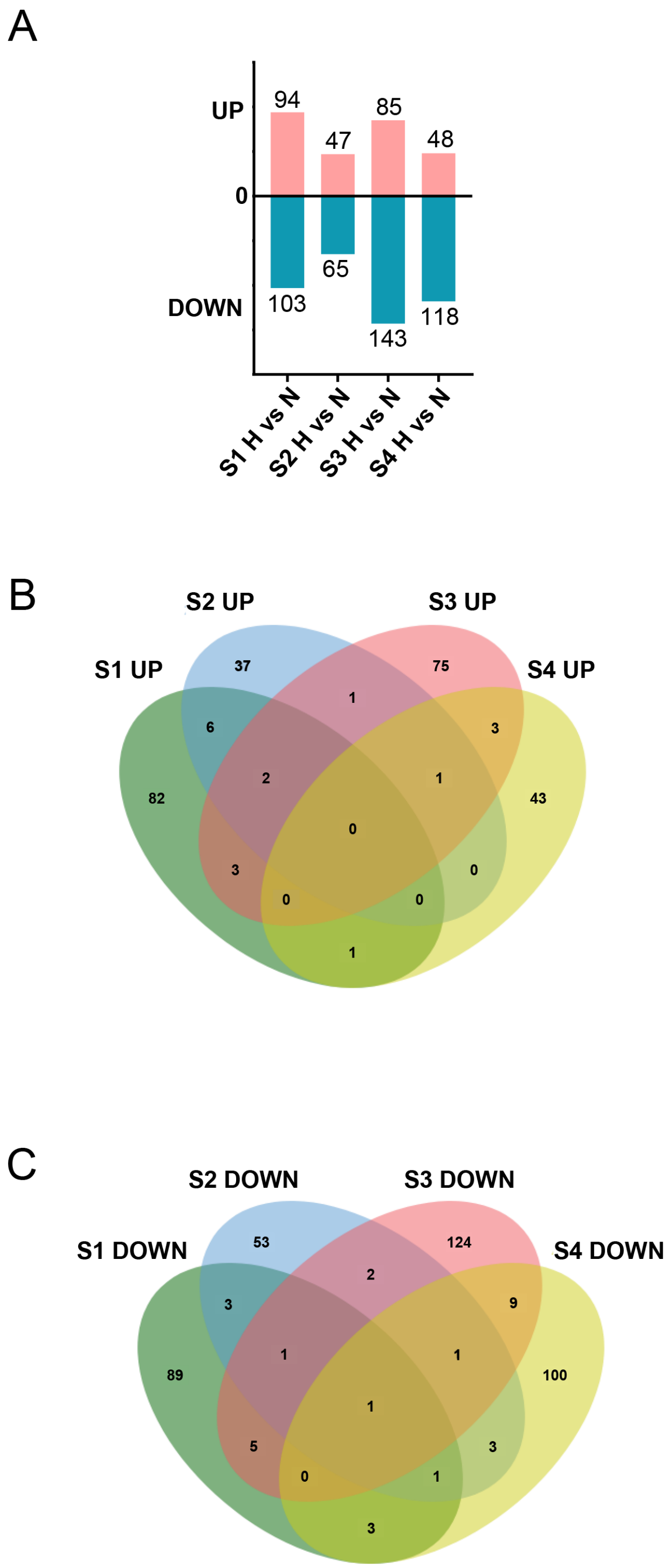

**Figure 5 Aggregation of DEGs after H treatment of cardiac fibroblasts from different age rats.** (A) Upregulated and downregulated DEGs between N and H treatment. (B) Venn diagram analysis of upregulated DEGs between two or more groups across the four timepoints. (C) Venn diagram analysis of downregulated DEGs between two or more groups across the four timepoints.

**Table 2 Up-regulated genes in different stages.**

| UP in stage | Gene | Name | Functions | GO term | KEGG |
| --- | --- | --- | --- | --- | --- |
| S1, S2, S3 | *Slco4a1* | Solute carrier organic anion transporter family, member 4a1 | Slco4a1 encodes OATP4A1, which is an uptake membrane transporter of metabolic products, such as steroid, thyroid and prostaglandin. High SLCO4A1 expression was connected to inflammation-associated pathways in high-grade serous ovarian cancer and thyroid cancer (DOI 10.3389/fphar.2022.946348; DOI 10.2147/IJGM.S339921) | | |
| | *Tgm1* | Transglutaminase 1 | TGase-1 is $Ca^{2+}$-dependent enzyme, which is expressed at epidermal keratinocytes in the skin, and stratified squamous epithelium of the upper digestive tract and in lower female genital tract. Normal transglutaminase-1 enzyme catalyzes N$\varepsilon$-($\gamma$-glutamyl) lysine crosslinking precursor proteins, such as loricrin and involucrin (DOI 10.3390/ijms23052506) | | |
| S2, S3, S4 | *Slc6a12* | $\gamma$-aminobutyric acid transporter 2(GAT2) | GAT-2 deficiency promoted the differentiation of naïve T cells into Th1 cells (DOI 10.3389/fimmu.2021.667136) is a transporter for the neurotransmitter GABA and osmolyte betaine. To date, most studies on BGT-1 have focused on its functions in the nervous system and renal osmotic homeostasis (DOI 10.1016/j.bbadis.2019.165634) | | |
| S1, S2 | *Paqr5* | Progestin and adipoQ receptor 5 | PAQR5 downregulation correlated with tumor stage, cancer grade, lymph node invasion, and distal metastasis only in clear cell RCC (ccRCC) tissues. TGFβ1 treatment significantly reduced PAQR5 gene expression in Immortalized Ovarian Surface Epithelial Cells (DOI 10.3389/fonc.2021.827344). | | |
| | *Cxcl1* | | | | |
| | *Spint1* | serine peptidase inhibitor, Kunitz type 1 | Hepatocyte growth factor activator inhibitors (DOI 10.3389/fendo.2021.665666) Epidermal differentiation and barrier function require well-controlled matriptase and prostasin proteolysis, in which the Kunitz-type serine protease inhibitor HAI-1 represents the primary enzymatic inhibitor for both proteases (DOI 10.1007/s13577-021-00488-1) | | |
| | *Nwd1* | NACHT and WD repeat domain-containing protein 1 | A member of STAND family, in the regulation of the assembly of a giant multi-enzyme complex that enables efficient *de novo* purine biosynthesis during brain development (DOI 10.1016/j.gep.2022.119284). *In vitro* experiments showed that knockdown of NWD1 inhibited dendritic growth and synaptogenesis (DOI 10.1016/j.neuropharm.2021.108919) | | |
| | Rn60_10_0890.5 | | | | |
| | AABR07005593.1 | | | | |

Table 2 (*continued*)

| UP in stage | Gene | Name | Functions | GO term | KEGG |
|---|---|---|---|---|---|
| S1, S3 | *Tfpi2* | Tissue factor pathway inhibitor 2 | Recent studies have shown that TFPI-2 translocates into the nucleus, where it modulates the transcription of the matrix metalloproteinase-2 (MMP-2) gene (DOI 10.1007/s10528-023-10340-w) glucose metabolism-related genes (glucose metabolism-related genes) can inhibit MMP1, MMP10 and the PI3K/AKT signaling pathway activity (DOI 10.1038/s41416-022-01831-5) | | |
| | *Il6* | | | | |
| | *Plekhs1* | Pleckstrin homology domain-containing S1, is a largely uncharacterised gene | PLEKHS1 has also been implicated as a potential mediator for the onset of T2DM in people with obesity. suggesting PLEKHS1 to be a potential mediator of the onset of T2DM in obese rats (DOI 10.3390/ijms222011150) | | |
| S1, S4 | *Plekhd1* | | | | |
| S2, S3 | *Serpinb2* | serpin family B member 2 | Monocytic marker (DOI 10.1158/2326-6066.CIR-22-0462) Moreover, the Serpinb2 expression level was increased in fibroblasts in fibrotic ECM, an inhibitor of extracellular urokinase plasminogen activator (DOI 10.1038/s42003-022-04333-5) | | |
| S3, S4 | *Nup210I* | nucleoporin 210 | | | |
| | *Plag1* | Pleomorphic adenoma gene 1 PLAG1 zinc finger | Upstream mediators such as PLAG1, can regulate AMPK-mediated metastasis (DOI 10.1016/j.lfs.2021.119649) The PLAG1 gene is best known as an oncogene associated with certain types of cancer, most notably pleomorphic adenomas of the salivary gland (DOI 10.1530/JOE-15-0449) PLAG1 overexpression is unable to transform fibroblasts with a targeted disruption of the insulin-like growth factor 1 receptor (IGF1R), required for IGF2 (but also IGF1) function | | |
| | *LOC498675* | | | | |

An examination of the 429 downregulated DEGs identified 28 genes in two or more groups (Fig. 5C, Table 3). Among these, one universally downregulated DEG across the four CF stages was identified: *Slc6a6* (Fig. 5C, Table 3). This gene encodes a taurine transporter that has been documented to play important roles in cardiac dysfunction and dilated cardiomyopathy (*Garnier et al., 2021*). The remaining 27 DEGs were primarily associated with maintaining pancreatic β cell function, cytoskeleton, and cell signaling transduction (Table 3). These results suggest that HG exerts multifaceted effects on CFs.

### Effects of HG treatment on gene expression in S1-CFs

In S1-CFs stimulated with HG, 197 DEGs were significantly altered, including 94 upregulated and 103 downregulated genes (Figs. 6A, 6B). KEGG pathway analysis and GO enrichment assays were performed on the upregulated DEGs. The top 20 enriched KEGG pathways (Fig. 6C) were predominantly associated with inflammation, such as the IL-17 signaling pathway, cytokine-cytokine receptor interaction, and TNF signaling

**Table 3 Down-regulated genes in different stages.**

| Down in stage | Gene | Name | Functions |
| --- | --- | --- | --- |
| S1, S2, S3, S4 | *Slc16a6* | Solute carrier family 16 member 6, encoding Monocarboxylate transporter 7 (MCT7) | In humans, MCT7 is primarily expressed in the liver, brain, and endocrine pancreas, transports ketone body, β-hydroxybutyrate, and taurine (DOI 10.1016/j.jbc.2022.101800) orthologous human gene locus SLC16A6 are highly significantly associated with human height (DOI 10.3389/fphys.2018.01936) We showed that induction of hepatic Slc16a6 in db/db mice was inhibited by Caloric restriction (DOI 10.1038/srep30111). After starvation or limited energy intake, the expression of Slc16a6 in the liver of obese mice decreased. Transport ketone body (DOI 10.1124/pr.119.018762) |
| S1, S2, S3 | *Bfsp1* | Beaded filament structural protein 1, filensin | This gene encodes a lens-specific intermediate filament-like protein named filensin, The encoded protein is expressed in lens fiber cells after differentiation has begun. This protein functions as a component of the beaded filament which is a cytoskeletal structure found in lens fiber cells. Mutations in this gene are the cause of autosomal recessive cortical juvenile-onset cataract. vimentin, BFSP1, and BFSP2 all belong to intermediate filament (IF) proteins, which belong to the cytoskeleton of fibroblasts. |
| S1, S2, S4 | *Smpd3* | sphingomyelin phosphodiesterase 3, ceramide biosynthesis enzymes | This gene belongs to cell migration and differentiation, metabolic function. It plays a role through ceramide, including oxidative stress, PP2A, PKCz, JNK and IKKβ pathway to promote the occurrence of metabolic diseases. (DOI 10.1111/obr.13248) |
| S2, S3, S4 | *Oprd1* | Opioid receptor Delta 1, belongs to GPCR proteins | OPRD1 encodes a receptor for enkephalins, which have been shown to both inhibit and stimulate insulin secretion (DOI 10.1172/JCI163612) |
| S1, S2 | *Esyt3* | extended synaptotagmin 3 | ESYT3 was expressed in human lymphocyte, vascular endothelial cell, and smooth muscle cell (DOI 10.1016/j.atherosclerosis.2011.06.017), $Ca^{2+}$-sensor proteins with multiple C2 domains |
| | *Syn1* | Synapsin-1 | Synapse-related protein, Synapsin-I (SYN1) is a presynaptic phosphoprotein crucial for synaptogenesis and synaptic plasticity, but why it is expressed in cardiac fibroblasts remains unknown. |
| | *LOC102553099* | | |
| S1, S3 | *Slc12a8* | Solute carrier family 12 member 8, | NAD(+) precursor transporters, the level is negatively correlated with aging (DOI 10.1016/j.celrep.2022.111131) |
| | *Map2k6* | mitogen-activated protein kinase kinase 6, | This protein phosphorylates and activates p38 MAP kinase in response to inflammatory cytokines or environmental stress. As an essential component of p38 MAP kinase mediated signal transduction pathway, this gene is involved in many cellular processes such as stress induced cell cycle arrest, transcription activation and apoptosis. |
| | *Ppp1r42* | protein phosphatase 1 regulatory subunit 42, | A leucine-rich repeat protein (PPP1R42) that contains a protein phosphatase-1 binding site and translocates from the apical nucleus to the centrosome at the base of the flagellum during spermiogenesis (DOI 10.1111/boc.201300019) |
| | *Sprn* | shadow of prion protein | The SPRN gene encodes the Shadoo glycoprotein (Sho), a central nervous system-expressed member of the prion protein superfamily |
| | *AC127756.1* | | |

Table 3 (*continued*)

| Down in stage | Gene | Name | Functions |
|---|---|---|---|
| S1, S4 | *Plcxd3* | phosphatidylinositol specific phospholipase C X domain containing 3 | Has been shown to influence pancreatic beta-cell function by disrupting insulin signaling (DOI 10.3390/genes11060665) |
| | *Nkapl* | NFKB activating protein like | |
| | *Cngb1* | cyclic nucleotide gated channel subunit beta 1 | In humans, the rod photoreceptor cGMP-gated cation channel helps regulate ion flow into the rod photoreceptor outer segment in response to light-induced alteration of the levels of intracellular cGMP. |
| S2, S3 | *LOC100910255* | | |
| | *AABR07068590.1* | | |
| S3, S4 | *Drp2* | dystrophin related protein 2 | |
| | *Fam198a* | golgi associated kinase 1A, | Fam198a is a member of four-jointed protein kinases, a secreted protein kinase family. It was identified as a caveolae-associated protein and colocalized with cavin-1 and caveolin-1 in both tissues and cells (DOI 10.1093/abbs/gmy105) |
| | *Clmn* | Calmin | was identified as one of all-trans retinoic acid (atRA)-responsive genes (DOI 10.2147/IJGM.S326960) |
| | *Cyp2d4* | | |
| | *Ednra* | endothelin receptor type A, | Fibroblast-related molecules, (DOI 10.3389/fonc.2022.808448) |
| | *Fgd2* | FYVE, RhoGEF and PH domain containing 2 | FGD2, a member of FGD family, contains a Dbl homology domain (DH) and two pleckstrin homology domains segregated by a FYVE domain. The DH domain has been deduced to be responsible for guanine nucleotide exchange of CDC42 to activate downstream factors (DOI 10.1016/j.pep.2020.105693) |
| | *Oas1k* | 2′-5′oligoadenylate synthetase 1K | Predicted to enable 2′-5′-oligoadenylate synthetase activity and double-stranded RNA binding activity. Predicted to be involved in defense response to virus and innate immune response. Predicted to be located in cytoplasm. Predicted to be integral component of membrane. |
| | *MGC109340* | | |
| | *LOC680693* | | |

pathway. GO analysis revealed that these upregulated genes were primarily involved in biological process (50.58%) (Fig. 6D). Additionally, 35.54% were associated with cell components and 13.88% with molecular functions (Fig. 6D). Specifically, 60 genes were implicated in cellular process, 55 in single-organism process, and 43 in metabolic process (Fig. 6D). Regarding cell components, 55 genes were involved in cell and cell part each, whereas 36 genes were involved in organelle and membrane each (Fig. 6D). Molecular function analysis indicated 52 genes associated with molecular binding and 19 with catalytic activity (Fig. 6D).

For the downregulated genes, KEGG analysis showed associations with the TNF signaling pathway, Toll-like receptor signaling pathway, Th17 cell differentiation (Fig. 6E). GO analysis revealed that these genes were implicated in biological processes (51.14%) (Fig. 6F), with 36.34% associated with cell components, and 12.52% associated with molecular functions (Fig. 6F). Specifically, 52 genes were engaged in cellular process, 49 in single-organism processes, and 41 in metabolic processes (Fig. 6F). Regarding cell

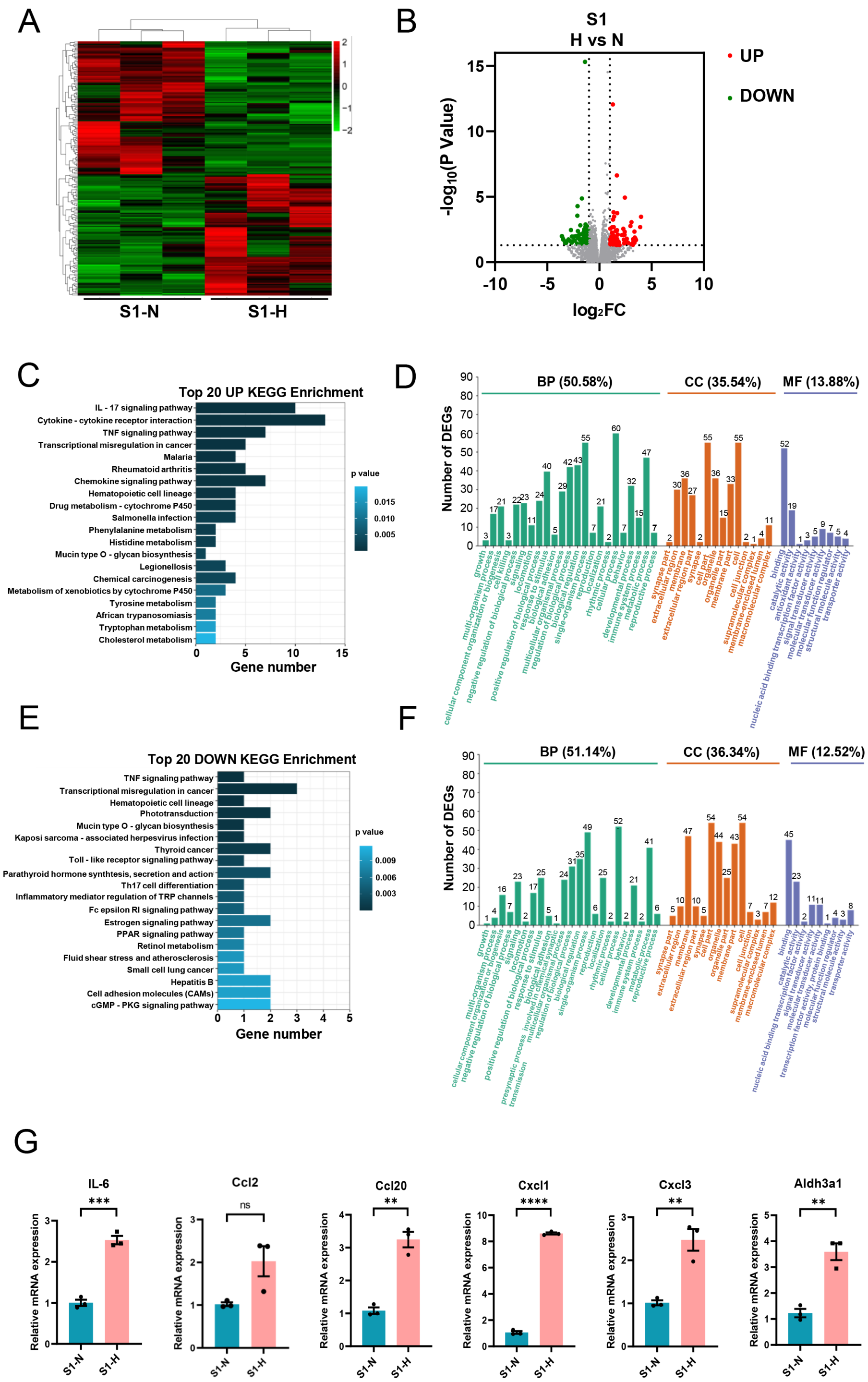

**Figure 6 Analysis of differentially expressed genes in S1 (0-3d) CFs after N and H treatment.** (A) Heatmap analysis showing DEGs between N and H treatment in S1. (B) Volcano plot with red dots representing upregulated DEGs and green dots representing downregulated DEGs; grey area shows genes without differential expression. 

**Figure 6 (…continued)**
(C) Top 20 enriched KEGG pathways for upregulated DEGs. (D) GO enrichment analysis for upregulated DEGs. (E) Top 20 enriched KEGG pathways for downregulated DEGs. (F) GO enrichment analysis for downregulated DEGs. (G) qRT-PCR analysis of *IL-6*, *Ccl2*, *Ccl20*, *Cxcl1*, *Cxcl3*, *Aldh3a1* mRNA levels in S1. For *IL-6*, *Ccl20*, *Cxcl1*, *Cxcl3* and *Aldh3a1*, *T*-test were used for statistical analysis, $^{****}P < 0.0001$; $^{***}P < 0.001$; $^{**}P < 0.01$; $^{*}P < 0.05$. *Ccl2* was not a normal distribution, and Mann–Whitney test was used for statistical analysis, ns, no significance.

components, 54 genes were involved in cell and cell part, 47 in membrane, and 44 in organelle (Fig. 6F). Molecular function analysis identified 45 genes involved in binding and 23 in catalytic activity (Fig. 6F).

Subsequently, we validated a subset of DEGs with significant fold-changes using quantitative real-time PCR, encompassing *IL-6*, *Aldh3a1*, *Ccl2*, *Ccl20*, *Cxcl1*, and *Cxcl3*. The results confirmed marked upregulation of these inflammatory factors in S1-CFs after HG treatment (Fig. 6G).

### Effects of HG treatment on gene expression in S2-CFs

After HG treatment of S2-CFs, 112 genes were significantly modified, comprising 47 upregulated and 65 downregulated genes (Figs. 7A, 7B). KEGG analysis of the upregulated genes revealed enrichment for pathways associated with cardiac muscle contraction, galactose metabolism, calcium signaling pathway, and AMPK signaling pathway among the top 20 pathways (Fig. 7C). In contrast to the S1 stage, the upregulated KEGG pathways in the S2 stage were associated with lipid and carbohydrate metabolism as well as signal transduction, indicating that HG challenge of S2-CFs primarily affects cell metabolic process. GO analysis demonstrated that these upregulated genes were primarily involved in biological processes (51.06%) (Fig. 7D). Additionally, 29.79% were associated with cell components and 19.15% with molecular functions (Fig. 7D). Specifically, 31 genes were involved in cellular process, 27 in single-organism process and positive regulation of biological process, and 26 in metabolic process (Fig. 7D). Regarding cell components, 30 genes were involved in cell and cell part, 24 in organelle, and 22 in membrane and membrane part each (Fig. 7D). Molecular function analysis identified 29 genes associated with binding and 10 with catalytic activity (Fig. 7D).

The downregulated KEGG pathways included type I diabetes mellitus, apart from cardiac muscle contraction and galactose metabolism (Fig. 7E). GO analysis revealed that these downregulated DEGs were primarily implicated in biological processes (45.23%) (Fig. 7F), with 32.01% linked to cell components and 22.76% to molecular functions (Fig. 7F). Specifically, 43 genes were involved in cellular process, 41 in single-organism process, and 32 in biological regulatory process (Fig. 7F). Regarding cell components, 42 genes were associated with cell and cell part each, 34 with membrane, and 32 with organelle (Fig. 7F). Molecular function analysis identified 39 genes involved in molecular binding and 17 in catalytic activity (Fig. 7F).

Moreover, we validated *Atp1b2* in S2-CFs, a gene markedly affected by HG treatment, using quantitative real-time PCR. The results showed that its expression was significantly upregulated following HG treatment (Fig. 7G).

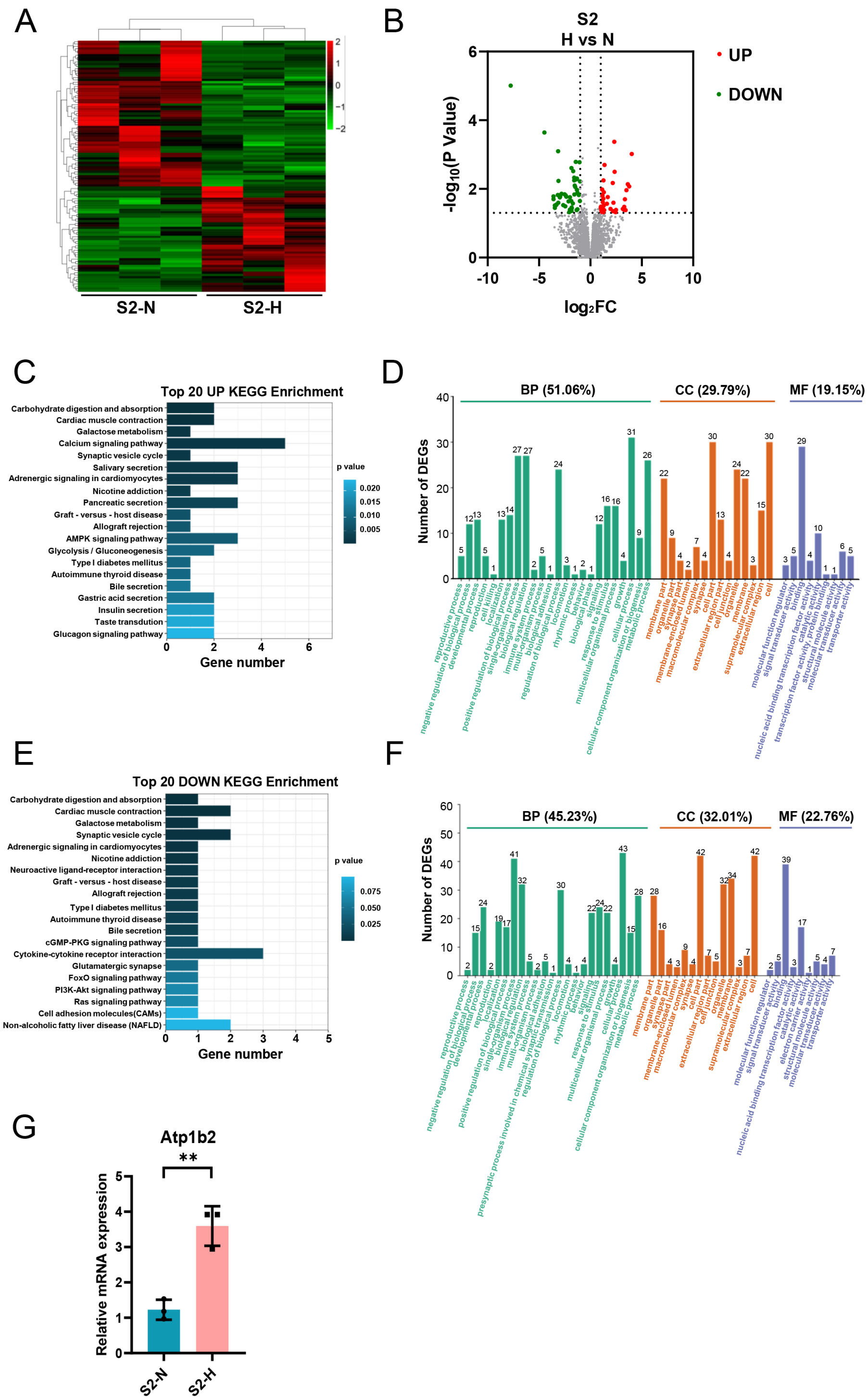

**Figure 7 Analysis of DEGs in S2 (3-4w) CFs after N and H treatment.** (A) Heatmap analysis showing DEGs between N and H treatment in S2. (B) Volcano plot depicting expression of DEGs. Red and green dots represent upregulated and downregulated DEGs, respectively. (C) Top 20 enriched KEGG pathways for upregulated DEGs. 

**Figure 7 (…continued)**
(D) GO enrichment analysis mapping upregulated DEGs to "Biological Process" (BP), "Cellular Component" (CC) and "Molecular Function" (MF) domains. (E) Top 20 enriched KEGG pathways for downregulated DEGs. (F) GO enrichment analysis mapping downregulated DEGs to "Biological Process" (BP), "Cellular Component" (CC) and "Molecular Function" (MF) domains. (G) qRT-PCR analysis of *Atp1b2* mRNA levels in S2. *T*-test were used for statistical analysis, $^{**}P < 0.01$.

### Effects of HG treatment on gene expression in S3-CFs

HG treatment of S3-CFs resulted in changes in 228 genes, of which 85 were upregulated, and 143 were downregulated (Figs. 8A, 8B). KEGG analysis showed that the upregulated DEGs were primarily involved in pathways such as calcium signaling and Wnt signaling (Fig. 8C). GO analysis demonstrated that these upregulated DEGs were mainly involved in biological processes (55.72%) (Fig. 8D). Additionally, 27.49% were related to cell components and 16.79% to molecular functions (Fig. 8D). Specifically, 57 genes were implicated in cellular process, 55 in single-organism process, and 48 in biological regulatory process (Fig. 8D). Regarding cell components, 53 genes were associated with cell and cell part, 38 with organelle, and 37 with membrane (Fig. 8D). Concerning molecular functions, 52 genes were associated with binding (Fig. 8D).

KEGG analysis of the downregulated genes revealed enrichment for pathways, including calcium signaling, the intestinal immune network for IgA production, and the Wnt signaling pathway. GO analysis showed that these downregulated genes primarily participated in biological processes (52.81%) (Fig. 8F), with 17.45% linked to cell components and 29.74% linked to molecular functions (Fig. 8F). Specifically, 87 genes were involved in cellular process, 75 in single-organism process, and 66 in biological regulatory process (Fig. 8F). Regarding cell components, 85 genes were involved in cell and cell part each, 66 in membrane, and 57 in organelle (Fig. 8F). Regarding molecular functions, 78 genes were involved in binding, and 33 in catalytic activity (Fig. 8F). Similarly, qRT-PCR was performed to validate the DEG of interest in S3, primarily *Lef1*.

### Effects of HG treatment on gene expression in S4-CFs

Analysis of RNA-seq data from S4-CFs treated with HG revealed 166 DEGs, with 48 upregulated and 118 downregulated genes (Figs. 9A, 9B). KEGG analysis demonstrated that the upregulated genes were mainly enriched in pathways such as the calcium signaling pathway, inflammatory mediator regulation of TRP channels insulin resistance, adipocytokine signaling pathway, and cAMP signaling pathway (Fig. 9C). GO analysis showed that these upregulated genes primarily participated in biological processes (58.03%) (Fig. 9D), with 20.41% affiliated with cell components, and 21.56% with molecular functions (Fig. 9D). Specifically, 30 genes were involved in cellular process, 26 in single-organism process, and 24 in biological regulatory process (Fig. 9D). For cell components, 23 genes were involved with cell and cell part each, 22 with membrane, and 19 with membrane part (Fig. 9D). Regarding molecular functions, 28 genes were linked to binding and 10 to catalytic activity (Fig. 9D).

KEGG analysis of downregulated genes showed enrichment in pathways such as tryptophan metabolism, calcium signaling pathway, cAMP signaling pathway, and valine,

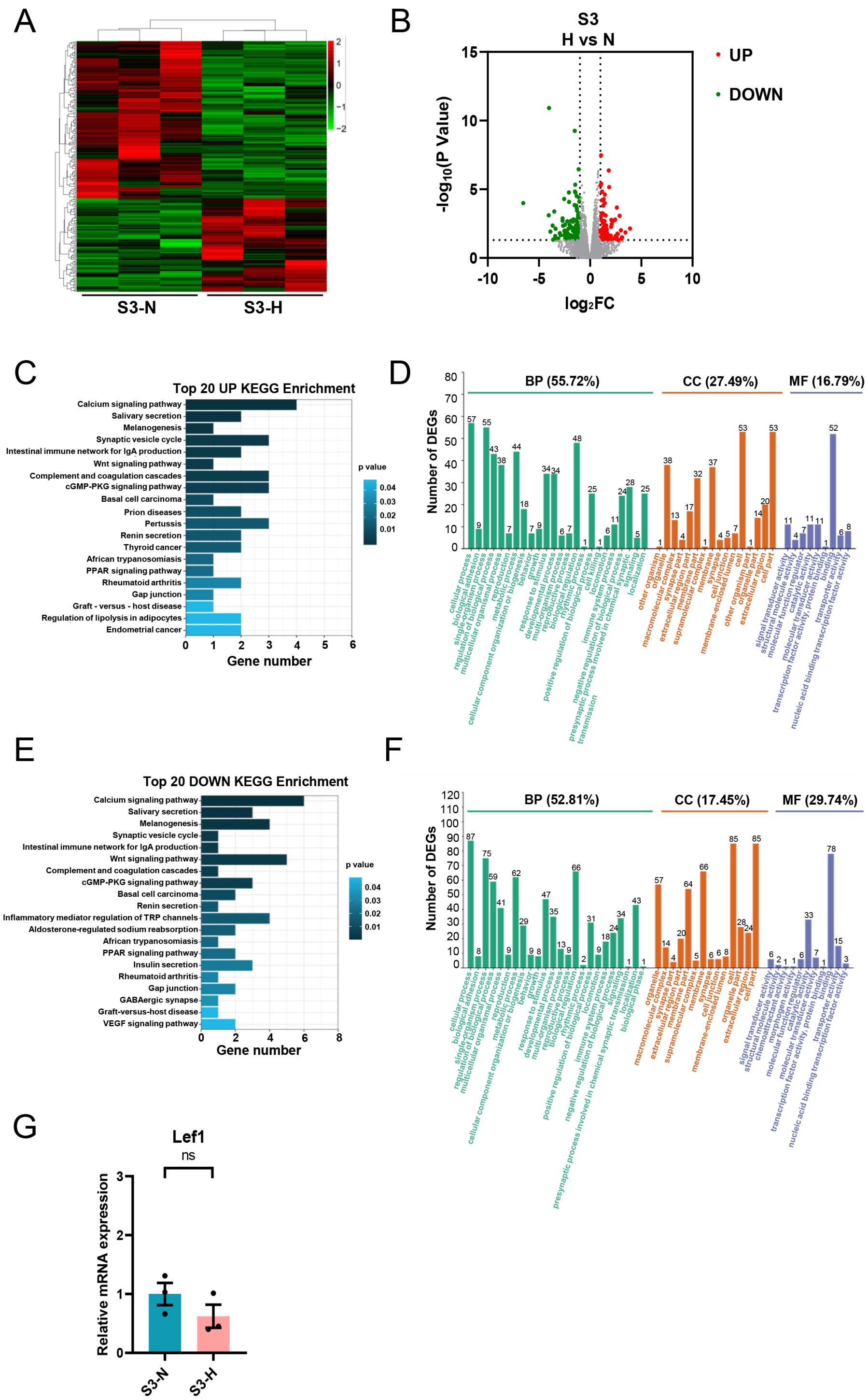

**Figure 8 Analysis of DEGs in S3 (10-13w) CFs after N and H treatment.** (A) Heatmap showing DEG expression after N and H treatment. (B) Volcano plot depicting upregulation and downregulation of DEGs in H *versus* N treatment. Red and green dots denote upregulated and downregulated DEGs, respectively. (C) Top 20 enriched KEGG pathways for upregulated DEGs. 

**Figure 8 (…continued)**
(D) GO analysis mapping upregulated or downregulated DEGs to "Biological Process" (BP), "Cellular Component" (CC) and "Molecular Function" (MF) domains. (E) Top 20 enriched KEGG pathways for downregulated DEGs. (F) GO analysis of DEGs between N and H treatment, displaying allocation to "Biological Process" (BP), "Cellular Component" (CC) and "Molecular Function" (MF) domains. (G) qRT-PCR analysis of *Lef1* mRNA levels in S3. *T*-test were used for statistical analysis, ns, no significance.

leucine and isoleucine degradation (Fig. 9E). This indicated that the effect of HG on aged CFs primarily manifests in their amino acid metabolism. GO analysis revealed that these downregulated genes were mainly involved in biological processes (53.33%) (Fig. 9F), with 31.11% linked to cell components and 15.56% to molecular functions (Fig. 9F). Specifically, 68 genes were implicated in cellular process, 68 in single-organism process, and 45 in metabolic process (Fig. 9F). Regarding cell components, 64 genes were associated with cells and cell parts each, 46 with membrane, 41 with membrane part, and 40 with organelle (Fig. 9F). Regarding molecular functions, 56 genes were associated with molecular binding and 29 with catalytic activity (Fig. 9F). Similarly, qRT-PCR was used to validate the DEGs of interest in S4, primarily *Erbb3*, *Prkcq*, and *Ddb2*.

### The mRNA expression of *ACTA2* and the differences in cell morphology under NG and HG conditions

We observed the RNA-seq results of *ACTA2* treated by NG and HG in four age groups, and found that the sequencing of *ACTA2* basically increased with age, indicating that CFs was activated with the increase of age (Fig. S1).

At the same time, we had attached photographs of CFs under NG and HG conditions for all age groups. It could be seen that under NG condition, with the increase of age, CFs gradually activated from static state, and the shape changed from flat to polygonal or star. Similar morphological changes occurred in CFs under HG condition, although the degree of activation in S4 stage decreased (Fig. S2).

## DISCUSSION

Under HG conditions, oxidative stress, inflammatory responses, neuroendocrine-immune responses, and growth factor release are enhanced, activating cardiac interstitial and vascular peripheral CFs. This leads to excessive deposition of extracellular matrix proteins, contributing to cardiac fibrosis and exacerbating morbidity and mortality in patients with DCM (*Frangogiannis, 2019*; *Tuleta & Frangogiannis, 2021a*; *Tuleta & Frangogiannis, 2021b*). CFs are pivotal in the pathogenesis of diabetic cardiac fibrosis. While most *in vivo* models exploring diabetic heart disease have utilized adult animals, *in vitro* experiments, have largely employed CFs derived from neonatal rats (*Lin et al., 2021*; *Liu et al., 2020*). However, whether neonatal rat CFs accurately represent the role in adult human diabetes remains to be validated owing to inherent species-specific characteristics. This study isolated CFs from neonatal, juvenile, adult, and aging rats and evaluated their gene expression changes and responses to HG stimulation.

Under NG treatment, 5,735 genes were altered in CFs, with 1,825 upregulated and 3,910 downregulated. With increasing age, 909 DEGs in clusters 2 and 10 exhibited upward

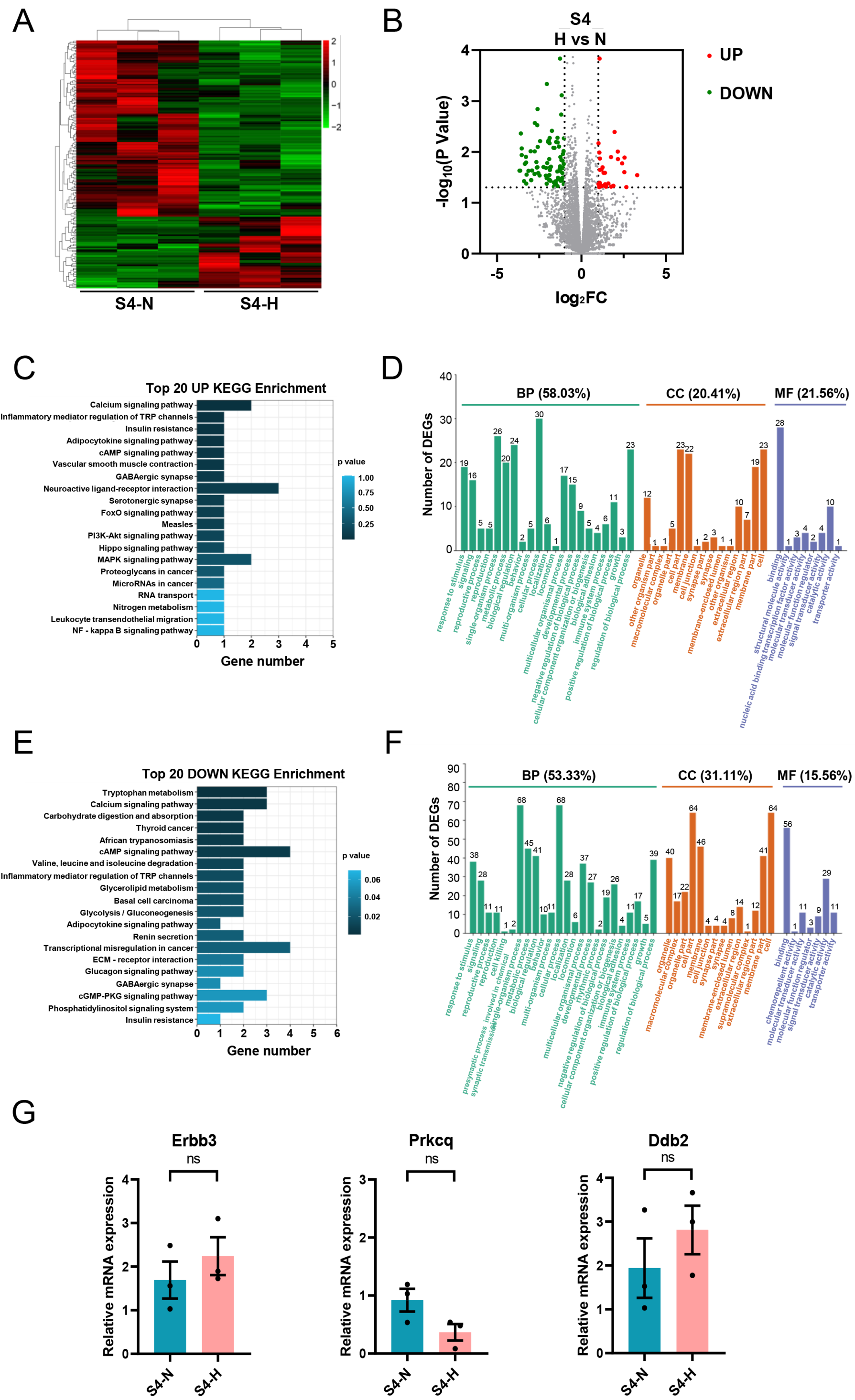

**Figure 9 Analysis of DEGs in S4 stage (20 m) CFs after N and H treatment.** (A) Heatmap showing DEG expression after N and H treatment. (B) Volcano plot depicting upregulation and downregulation of DEGs in H *versus* N treatment. 

**Figure 9 (…continued)**
(C) Top 20 enriched KEGG pathways for upregulated DEGs. (D) GO analysis mapping upregulated or downregulated DEGs to "Biological Process" (BP), "Cellular Component" (CC) and "Molecular Function" (MF) domains. (E) Top 20 enriched KEGG pathways for downregulated DEGs. (F) GO analysis of DEGs between N and H treatment, displaying allocation to "Biological Process" (BP), "Cellular Component" (CC) and "Molecular Function" (MF) domains. (G) qRT-PCR analysis of *Erbb3*, *Prkcq*, *Dbdh2* mRNA levels in S4. *T*-test were used for statistical analysis, ns, no significance.

trends, whereas 1,602 DEGs in clusters 3, 5, and 7 showed downward trends. After HG treatment, CFs at different stages showed varying responses. The DEGs in S1-CFs were mainly involved in inflammatory signaling pathways, whereas those in S2-CFs primarily participated in cardiac muscle contraction and galactose metabolism. The DEGs in S3-CFs were chiefly implicated in the calcium and Wnt signaling pathways, whereas the DEGs in the S4-CFs were mostly enriched in the calcium signaling and amino acid metabolic pathways.

## Differences in gene expression of rat CFs across four stages

Our analysis revealed differences in gene expression related to fibrosis, inflammation, oxidative stress, and the cell cycle among CFs at different stages.

### *Cell proliferation and senescence*

Aging exerts an irreversible impact on the development of cardiovascular diseases, with morbidity increasing as age advances (*Cianflone et al., 2020*). In cardiac tissues, fibroblasts rank second only to cardiomyocytes and play crucial roles in cardiac structure and function. However, the proliferative capacity of fibroblasts decreases with age, diminishing their ability to repair cardiac tissues after injury. Our findings indicate that, under NG condition, the expression of cell proliferation-related genes involved in DNA replication and chromosome separation, such as *Mcm2*, *Mcm7,* and *Mcm8*, markedly decreases at the S4 stage, accelerating the senescence of cardiac tissues.

### *Inflammation, oxidative stress and cardiac hypertrophy*

In cardiac tissues, the death of cardiomyocytes triggers local immune cells, initiating pro-inflammatory cascades that generate cytokines and chemokines. These molecules, in turn, activate CFs, prompting them to secrete procollagen and other ECM molecules. This process increases the production of pro-inflammatory cytokines, chemokines, and growth factors (*Díez, González & Kovacic, 2020*). This indicates that myocardial inflammation and fibrosis often coexist in nonischemic heart failure models (*Bacmeister et al., 2019*). Oxidative stress is a double-edged phenomenon. Appropriate oxidative stress facilitates cell growth and repair. However, it can also upregulate TGF-β1 expression, promoting fibroblast transformation and collagen synthesis, thereby leading to cardiac fibrosis (*De Geest & Mishra, 2022*). Studies have demonstrated that inhibiting ROS signaling in cardiac tissues can reduce myocardial remodeling and relieve myocardial fibrosis (*Siwik & Colucci, 2004*). Previous evidence highlights the critical roles of β-MHC encoded by *Myh7* in cardiac contractile function. Similarly, *Nppa* regulates vascular dilation and suppresses cardiac remodeling and hypertrophy. Although these two genes are mainly expressed in

cardiomyocytes, they are also present in CFs, potentially contributing to the maintenance of cardiac function.

The expression levels of specific pro-inflammatory cytokines (*IL-33*, *Tgfb3*) and chemotactic factors (*Cxcl13* and *Cxcl14*) displayed an age-related upregulation. Furthermore, oxidative stress-regulated genes (*Sod3* and *Gpx3*) were enriched at the S4 stage. These findings indicate that the robust induction of inflammation-related cytokines and chemokines, along with exacerbated oxidative stress responses, collectively contribute to the fibrogenic cascade. Additionally, we observed an attenuation of the cardiomyopathy-linked genes *Nppa*, *Myh6*, *Myh7*, and *Actc1*, indicating decreasing trends in cardiac contraction and relaxation functions with advancing age.

#### *Effect of aging on matrix deposition and turnove*

Collagen expression is regulated by the TGFβ1 signaling pathway, which is critical in the development of fibrosis (*Frangogiannis, 2020*; *Meng, Nikolic-Paterson & Lan, 2016*; *Peng et al., 2022*). The binding of TGFβ1 to the cell membrane surface receptors TRβ1-TRβ2 triggers alterations in downstream gene expression (*Frangogiannis, 2020*), such as α-SMA, thereby promoting fibrosis. Under NG condition, we observed that the expression of α-SMA showed an overall increasing trend with advancing age. Similarly, the expression of *Col1a1* and *Col3a1* also exhibited an overall upward trend from stages S1 to S4. Additionally, *Col8a1*, *Col9a3*, *Col10a1*, *Col11a1*, *Col11a2*, *Col19a1* and *Col20a1* also exhibited an upward expression trend with age. This aligns with the expression trend of TGFβ1, revealing a pattern of enhanced fibrotic state with increasing age.

### Effects of HG on rat cardiac fibroblasts

#### *Cell proliferation and senescence*

Diabetes accelerates the progression of age-related pathological changes. Previous studies have shown that patients with diabetes are more prone to developing age-related complications, such as Alzheimer's disease and cardiovascular disease (*Biessels & Despa, 2018*; *Palmer et al., 2019*). In cardiac tissues, diabetes induces DNA damage, decreases DNA replication and cell differentiation capabilities, promotes cellular senescence and apoptosis, and ultimately results in the expansion of fibrotic regions in the myocardium (*Balakumar, Maung & Jagadeesh, 2016*). Our study showed that HG treatment of aging CFs from the S4 group downregulated cell proliferation-associated genes compared to NG conditions, exacerbating the impaired proliferative potential of senescent CFs to aggravate DCM.

#### *Inflammation, oxidative stress and cardiac hypertrophy*

Previous studies have shown that HG increases the production of pro-inflammatory cytokines, directly promoting inflammatory activity (*Wang et al., 2020*). In addition, diabetes activates the NLRP3 inflammasome, thereby increasing systemic inflammation (*Vandanmagsar et al., 2011*). In cardiac tissues, activated NLRP3 inflammasomes serve as novel molecular markers for DCM, enhancing myocardial cytokine production and promoting macrophage infiltration, thereby exacerbating myocardial inflammation (*Arnold et al., 2018*; *Jia, Hill & Sowers, 2018*; *Kawaguchi et al., 2011*; *Wen, Ting & O'Neill, 2012*). Moreover, DCM leads to abundant secretion of chemokines that induce

cardiac inflammation (*Corinaldesi et al., 2021*; *Packer, 2018*; *Westermann et al., 2009*). Dysregulation of the inflammatory response can cause myocardial injury, promote myocardial fibrosis, and ultimately lead to progressive left ventricular functional impairment and adverse left ventricular remodeling (*Mann, 2015*). Our study indicated that HG treatment at the S1 and S3 stages increased the expression of the inflammatory cytokines (*IL-6*, *Tgfb2,* and *Tgfb3*) and chemokines (*Cxcl13*), indicating that HG stimulation heightened the tissue inflammatory response to accelerate the progression of cardiac fibrosis.

Oxidative stress plays a pivotal role in the development of DCM (*Arnold et al., 2018*). Various studies have shown that ROS can induce DNA damage, especially mitochondrial DNA damage, leading to excessive superoxide production and oxidative myocardial injury, thereby exacerbating DCM (*Bugger & Abel, 2014*; *Taye, Abouzied & Mohafez, 2013*). Additionally, ROS contributes to ventricular remodeling, with NADPH oxidase being a significant source of ROS in cardiovascular cells. Elevated ROS production promotes cardiac fibrosis, and contributes to the development of LV interstitial fibrosis (*Huynh et al., 2014*). Our results showed that after HG induction, the expression levels of superoxide dismutase (*SOD2*) and glutathione peroxidase (*GPx2*) were lower across all four stages compared to those in NG-treated CFs, indicating weakened antioxidant capacity, consistent with previous studies.

Previous studies have underscored the critical role of hyperglycemia in the pathogenesis of DCM. Hyperglycemia contributes to pathological hypertrophy and fibrosis of the myocardium, impairing contractile and diastolic cardiac functions. Our results showed decreased expression of cardiomyopathy-associated genes, such as *Nppa*, *Myh6*, *Myh7*, *Tnni1* and *Tnni3*.

### Collagen degradation and synthesis

Similar to NG conditions, collagen expression increases with aging under HG conditions. High glucose can increase TGFβ1 expression (*Tian et al., 2021*; *Zhang et al., 2016*), which also increases with age under HG. TGFβ1 alters the fibrotic state by changing the expression of downstream α-SMA. In our sequencing results, α-SMA expression also increased with age under HG, promoting an increase in collagen expression, such as *Col1a1*, *Col3a1*, *Col8a1*, *Col9a3*, *Col10a1*, *Col11a1*, *Col11a2*, *Col19a1*, and *Col20a1*, which can have detrimental effects on heart function (*Gil-Cayuela et al., 2016*; *Lopes et al., 2013*; *Sadri et al., 2022*; *Song et al., 2022*).

## Comparison of *ACTA2* expression under NG and HG conditions

In this study, we observed the RNA-seq results of *ACT2* treated with NG and HG in different age groups. The results showed that the expression of *ACTA2* increases with age, which indicates that CFs were activated with age. This is similar to the changes of transcription module observed in the aging process of human skin fibroblasts (*Lee & Shivashankar, 2020*). *ACTA2* activates the activation of fibroblasts in the process of aging and promotes the progress of fibrosis.

At the same time, we also attached CFs photos of all age groups under NG and HG conditions. Fibroblasts in a static state show a flat morphology (*Pirri et al., 2023*), and

they show polygons or stars in the activated state (*Ireland & Mielgo, 2018*). We observed that with the increase of age, CFs changed from inactive fibrous form to activated form (the activation degree of S4 CFs decreased under HG condition), which confirmed the activation of CFs by age. This may be related to the increase of *ACTA2* expression. Under the action of *ACTA2,* static CFs becomes active, have contractile ability, and secretes ECM, thus promoting the increase of fibrosis level (*Grigorieva et al., 2024*; *Wu, Zhan & Wang, 2024*).

It is imperative to note that our study did not involve direct *in vivo* experimentation with diabetic mice. Currently, the majority of experimental diabetes research employs diabetic rats; however, the CFs utilized in extensive *in vitro* experiments on diabetic cardiopathy do not correspond to the age of onset for T1DM and T2DM. Consequently, to investigate the alterations in gene expression within CFs across various age groups and the impact of high glucose concentrations on these cells at different ages, we exposed CFs from rats of varying ages to NG or HG conditions *in vitro*, and monitored differences in gene expression. Future studies might consider isolating primary CFs from T1DM and T2DM animal models to examine the discrepancies in CF behavior under disease conditions.

Although we have validated the gene expression data through qPCR, our conclusions may be subject to certain limitations, as our analysis was confined to primary CFs *in vitro*, without the inclusion of *in vivo* animal experiments. The validity and reliability of these observations necessitate further confirmation through animal studies to more thoroughly comprehend their potential clinical significance.

## CONCLUSION

In summary, transcriptomic analysis of CFs specific to different developmental stages showed the upregulation of inflammation- and fibrosis-associated genes alongside a contrasting downregulation of metabolic genes with aging. HG treatment led to age-dependent variations in these responses. These findings offer valuable insights into how CF functionality evolves over time, shedding light on the mechanisms underlying diabetic cardiac fibrosis and providing crucial experimental evidence for developing therapies targeting this disease.

### Funding

This work was supported by the Basic and Applied Basic Research Project of Guangdong Province of China (2024A1515013218), the Research team project of Prevention and Treatment of Diabetic Cardiomyopathy with Integrated Chinese and Western medicine (2024ZZ06), the Medical Science and Technology Research Fund of Guangdong Province (A2021311), and the Open Project of NHC Key Laboratory of Assisted Circulation and Vascular Diseases (KF202101). The funders had no role in study design, data collection and analysis, decision to publish, or preparation of the manuscript.

## Grant Disclosures

The following grant information was disclosed by the authors:

Basic and Applied Basic Research Project of Guangdong Province of China: 2024A1515013218.

Research team project of Prevention and Treatment of Diabetic Cardiomyopathy with Integrated Chinese and Western medicine: 2024ZZ06.

Medical Science and Technology Research Fund of Guangdong Province: A2021311.

Open Project of NHC Key Laboratory of Assisted Circulation and Vascular Diseases: KF202101.

## Competing Interests

The authors declare there are no competing interests.

## Author Contributions

- Quqian Mo conceived and designed the experiments, performed the experiments, analyzed the data, prepared figures and/or tables, authored or reviewed drafts of the article, and approved the final draft.
- Angyu Zhan conceived and designed the experiments, performed the experiments, analyzed the data, prepared figures and/or tables, authored or reviewed drafts of the article, and approved the final draft.
- Ruining Bai performed the experiments, prepared figures and/or tables, and approved the final draft.
- Shaoling Lin performed the experiments, analyzed the data, prepared figures and/or tables, and approved the final draft.
- Jiaojiao Feng performed the experiments, prepared figures and/or tables, and approved the final draft.
- Tongjun Li conceived and designed the experiments, performed the experiments, analyzed the data, prepared figures and/or tables, and approved the final draft.
- Zijian Lao conceived and designed the experiments, authored or reviewed drafts of the article, and approved the final draft.
- Xiao Yang conceived and designed the experiments, authored or reviewed drafts of the article, and approved the final draft.
- Keke Wang conceived and designed the experiments, authored or reviewed drafts of the article, and approved the final draft.
- Xianglu Rong conceived and designed the experiments, authored or reviewed drafts of the article, and approved the final draft.
- Lexun Wang conceived and designed the experiments, prepared figures and/or tables, authored or reviewed drafts of the article, and approved the final draft.

## Animal Ethics

The following information was supplied relating to ethical approvals (i.e., approving body and any reference numbers):

This study was approved by the Animal Care and Ethics Committee of Guangdong Pharmaceutical University (approval number: gpdulac17).

## Data Availability

All raw data are available at NCBI Gene Expression Omnibus (GEO): GSE269896.

## Supplemental Information

Supplemental information for this article can be found online at http://dx.doi.org/10.7717/peerj.19040#supplemental-information.

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
