# Peer review of "Effect of high glucose on the gene expression profiling in cardiac fibroblasts from rats at different ages"

_PeerJ, doi:10.7717/peerj.19040_

## Round 0.1 · original submission · Major Revisions

Two reviewers highlight some issues which you should consider and adjust your paper accordingly.

The Major Revision decision is based mainly on the issues surrounding the sample size and validation of the observed changes. You must justify your decision to use this small size with a carefully laid out power calculation and clearly explain the numbers of biological and technical replicates performed for the analysis.

I agree with the comments of reviewer-2 on the figure number and quality. Please address this.

Reviewer 1 ·

Basic reporting

no comment

Experimental design

no comment

Validity of the findings

no comment

Additional comments

The authors of this study isolated cardiac fibroblasts (CFs) from neonatal, juvenile, adult, and aging rats and evaluated their gene expression changes and responses to HG stimulation. This is an interesting study, since as the authors mention, there is no established in vitro fibroblast model which would reflect in the best way the fibroblast changes under in vivo diabetic conditions. I have some comments/questions to authors:
- Page 6, Methods: “T tests were used throughout the study and P-value < 0.05 was considered statistically significant”. Were all the results distributed normally, so that the T test was the only test used?
- Results: Did the authors observe any morphological changes or increased expression of markers such as aSMA or periostin indicating transformation of fibroblasts into myofibroblasts under normal or high glucose conditions?
- Discussion: The authors write that DEGs in clusters 2 and 10 which mainly included collagens were upregulated with aging. The effects of aging on the matrix deposition and turnover are not discussed in the first part of Discussion connected to aging. However, in the Discussion part connected to the effects of HG on matrix, the authors write that “with increasing age, collagen production declines, whereas intracellular protein degradation rates increase, thereby reducing collagen turnover and accelerating tissue aging”, which is actually the opposite of what they showed. Please comment on this and please create a new section devoted to the effects of aging on collagen turnover. Moreover, the authors write: “we observed varying expression levels of 34 collagen subtypes, with an initial increase in Col1a1 and Col3a1 levels followed by a decline in the S4 stage, likely due to degradation mediated by matrix metalloproteinases, such as Mmp2 and Mmp9(Guo & Piacentini 2003). Does this sentence refer to the effects of HG on fibroblasts? So how does HG change the actions of fibroblasts in terms of matrix remodeling in different stages of fibroblast development in comparison to normal conditions?
Page 5, Methods: “Principal component analysis analysis and power analysis calculation” -> analysis analysis, please correct this.
Page 7, Results: “Further analysis of these 968 commonly altered DEGs revealed that 167 were upregulated and 801 were downregulated (Fig. 3A)”. In the Fig. 3A 802 genes were downregulated. Please correct this.

Reviewer 2 ·

Basic reporting

The manuscript investigated the change in gene expression of cardiac fibroblasts isolated from rat at different ages after culturing and exposure to normal and high glucose concentrations. This is an interesting study which collected a lot of transcriptomics data and validated by q-PCR. However, there are several limitations to draw such conclusions as explained in the next section under experimental design.

Overall, this is a well written manuscript- but need to be more focused, it contains a lot of data that is not easy for the reader to grasp.

Abstract: results section needs to be sharpened a little bit; the key messages need to be clear and concise.

Experimental design

1- The cause of myocardial fibrosis in diabetic cardiomyopathy are not fully known. It is likely to be driven by various factors, including hyperglycemia, oxidative stress and inflammation. While the authors claim they wants to investigate diabetes-associated cardiac fibrosis, the cellular model with high/low glucose induction does not reflect or recapitulate the pathological cause of condition. I wonder why isolated fibroblasts are not taken from animals that developed diabetes?
2- It is not clear how biological replicates were obtained from the same rat? Or from three different rats? The statistical power calculation is not clear how the authors determined 3 replicates are enough for such RNA-seq experiment. Based on literature, 6 replicates are at least required for best performance of the data analysis tools (edgeR and DESeq2).
3- The transcriptomic comparison approach is not comprehensive that resulted in large number of figures. Why the authors did not look at the longitudinal changes (s1-s4) rather than in each stage. Figures 6-9 are repetitive, it would be useful, if there is a way to present the longitudinal changes over S1-S4 in one figure that can substitute Figures 6-9.

Validity of the findings

Validation was done by q-PCR only, there is no in vivo experiments.

Additional comments

Minor points: RNA was harvested, this was mentioned twice in the abstract and under method section- I assume the authors means RNA was extracted. Figures are blurry, need higher resolution to read the text/name of genes- Also statistics to reflect/define significance level are not added (figure 6).

·

Basic reporting

The authors present their work on the differential expression of genes from rat primary cardiac fibroblasts isolated at four different ages. Additional studies evaluated the effect of high glucose culture conditions on the gene expression profile from cardiac fibroblasts isolated from the four different age groups.

The writing and grammar is solid with not major concerns. The authors provide context and rationale for their studies. Figures are well labeled and clearly presented. Raw data has been provided by the authors, and their sequencing data has been uploaded into GEO for future public access.

Experimental design

The design appears sound and experimental details necessary for potential replication of these studies by other investigators are included.

The manuscript/study is relevant to PeerJ and should be of interest to readers.

Sequencing data were validated using qPCR approach.

No major concerns.

Validity of the findings

Overall the data are well presented and appear to have sufficient rigor and replication.

The discussion of findings is somewhat brief, but does provide context for the findings and provides readers with a launching point if they would like to dig further into the potential significance of specific differentially expressed genes.

Conclusions are within the bounds of the data presented and do not overreach.

Overall, I found the manuscript easy to follow and understand, and the data set should be useful for investigators interested in cardiac fibroblast and cardiac research in general. Additionally, the assessment of high glucose versus normal glucose culture conditions broadens the relevance of the study to include diabetes and other metabolic disorders.

No concerns.

Additional comments

I found this manuscript to be solid overall. While not particularly ground breaking in subject-matter or methodology, the results should prove quite useful for other researchers in the area of cardiac pathogenesis and fibroblast function.

---

## Round 0.2 · Minor Revisions

I am largely satisfied with your responses to the issues raised. However, I note that some of these issues have not been translated into changes within the paper. Thus, could I ask you to revise your paper again please, introducing discussion of the points raised below, and the new data you provided in the rebuttal.

Specifically:
Reviewer-1
#2 Explicitly state which datasets were not a normal distribution in the legend to the figure.
#3 Include data from this point and discuss the morphology point raised.

Reviewer-2
#2 Clear discussion of the limitations of no isolated fibroblast work should be noted.
#3 Animal numbers. No issue with your explanation, but this needs to be clearly stated in the Methods please.
#5 This limitation must be clearly stated.

Thanks

Reviewer 1 ·

Basic reporting

The manuscript is written in a well-structured and logical way, including introduction, methods, results, conclusions, and references.

Experimental design

The experimental design is sufficiently explained, the experiments are well conducted and exlained.

Validity of the findings

All data are provided, the statistical analysis is sound, and the conclusions are based on results.

Additional comments

The authors addressed all my comments. I have no further questions.

---

## Round 0.3 · accepted · Accept

Thank you for attending to these final comments.